## Research Article

primary care; brief interventions; early childhood; maternal; digital health

**Corresponding author:**
Victoria Binda;
Email: victoria.binda@gmail.com

# Group videoconferencing intervention "C@nnected" to enhance maternal sensitivity in primary care in Chile: A feasibility trial

Victoria Binda[1] , Marcia Olhaberry[2], Carla Castañon[1], Constanza Abarca[2], Catalina Caamaño[2] and Javier Moran-Kneer[3]

[1]Department of Family Medicine, Medicine Faculty, Pontificia Universidad Católica de Chile, Santiago, Chile; [2]School of Psychology, Social Sciences Faculty, Pontificia Universidad Católica de Chile, Santiago, Chile and [3]Center for Translational Studies in Stress and Mental Health (C-Estres), Universidad de Valparaíso, Valparaíso, Chile

## Abstract

Early interventions supporting parental sensitivity have proven effective. Despite advancements in telemedicine, research on remote group parenting interventions remains limited. This study evaluated the feasibility and acceptability of "C@nnected," a brief group videoconferencing intervention aimed at enhancing maternal sensitivity in mother–infant dyads in primary care settings in Santiago, Chile. A feasibility randomized controlled trial (RCT) was conducted using quantitative and qualitative methods. Of 44 mother–infant dyads randomized, 26 were assigned to receive the intervention, whereas 18 were allocated to the control group. Eligibility and recruitment rates were 89% and 36%, respectively, with adherence at 50% and follow-up at 64.5%. The intervention demonstrated high acceptability in both the quantitative and qualitative evaluations. Mothers who participated in the intervention showed high scores in credibility and expectancy and reported increased knowledge, stronger bonds with their children and greater satisfaction and competence in their motherhood role. This pilot study underscores the potential of "C@nnected" while identifying areas for improvement. The findings provide valuable insights into refining and further evaluating its efficacy through an RCT.

## Impact statement

This feasibility trial evaluated the feasibility and acceptability of "C@nnected," a brief group videoconferencing intervention designed to enhance maternal sensitivity among mother–infant dyads attending primary care in Chile. Our findings reveal high levels of acceptability, supported by both quantitative and qualitative data. Adherence rates reached 50%, comparable to those of similar in-person group parenting interventions. The intervention is easy to replicate, locally developed, culturally adapted to the target population and improves access to early parenting interventions. Key barriers, facilitators and recommendations were identified to refine the intervention design before conducting a randomized controlled trial to evaluate its effectiveness.

## Introduction

Ensuring that children reach their maximum development potential has become a priority of public policies (Richter et al., 2017). There is compelling evidence about the importance of maternal sensitivity for achieving adequate well-being, and cognitive and emotional development of children (Dagan et al., 2021). Maternal sensitivity, defined as the ability to detect and respond appropriately to the child's needs (Crittenden, 2005), is the key characteristic of interactions promoting the attachment security of the child (Bretherton, 2013). The more sensitive, responsive, attentive and cognitively stimulating the mother is, the better the results for the child, making the sensitivity and quality of the mother–child interaction one of the most relevant predictors of child development (Black et al., 2017).

The main risk factor associated with the presence of low maternal sensitivity is maternal depression (Stein et al., 2014; Binda et al., 2019). Perinatal depression poses important challenges for public health; the prevalence of perinatal depression is as high as 25% (Fan et al., 2024). A systematic review shows Chile as the country evaluated with the highest prevalence of postpartum depression (Hahn-Holbrook et al., 2018). Perinatal depression negatively impacts the quality of mother–infant interactions, which can interfere with children's development and mental health (Stein et al., 2014; Fan et al., 2024). In recent years, the significance of perinatal anxiety symptoms has gained attention, given that they have also been shown to negatively impact child development outcomes (Della Vedova et al., 2023).

The COVID-19 pandemic negatively impacted maternal mental health and significantly limited access to treatments, due to confinement measures and the overload of health services (Davenport et al., 2020). A prevalence study conducted in nine countries (Brazil, Chile, Cyprus, Greece, Israel, Portugal, Spain, Turkey, and the United Kingdom) in 2020 reported prevalence rates of 26% for depression, 20% for anxiety, and 15% for co-morbidity in pregnant women. In postpartum women, the prevalence rates were 32.7%, 26.6%, and 20.3%, respectively. These figures are significantly higher than those reported before the COVID-19 pandemic (Mateus et al., 2022). Another key factor associated with optimal early childhood development is postnatal bonding, which refers to parents' subjective experiences and emotional connection with their infants during the first year of life. The quality of maternal–infant bonding is important, as it is predictive of maternal sensitivity (Mass et al., 2015) and is related to child developmental outcomes: low levels of postnatal bonding have been linked to delayed socioemotional development in children, executive functioning difficulties and increased externalizing behaviors (Riera-Martín et al., 2018). Improving sensitivity and postnatal bonding, especially in mothers with postpartum depression, is essential to achieve comprehensive child development.

There is strong evidence supporting the effectiveness of early interventions in promoting healthy child development. A recent systematic review (Jeong et al., 2021) on parenting interventions targeting children in their first 3 years of life demonstrated significant improvements in early cognitive, language, motor and socioemotional development, as well as in attachment. These interventions also reduced behavioral problems and enhanced parenting knowledge, practices and parent–child interactions. The effects were notably greater among families with low socioeconomic status and when the interventions included components focused on responsive caregiving, yielding nearly four times the impact compared with interventions without this content. However, no clear evidence was found to indicate that the effectiveness of the interventions varied based on the child's age, intervention duration, delivery method or setting.

Telemedicine has developed strongly in recent years, boosted by the growing need for this type of care during the COVID-19 pandemic, especially in the field of mental health (Ohannessian et al., 2020). In Latin America, an increase in the use of Internet-based mental health interventions has also been observed, mainly caused by the difficult access to in-person mental health services and greater access to the Internet in the general population (Martínez et al., 2018). In the post-pandemic era, they are proposed as a way to improve access, prevent possible infections and reduce costs compared with in-person interventions (Tongseiratch et al., 2020). Recent studies show the effectiveness of telemedicine interventions to reduce postpartum maternal depressive symptoms (Zhao et al., 2021). In addition, a meta-analysis (Spencer et al., 2020) demonstrated the effectiveness of online parenting programs, most of which were self-guided and lacked a synchronous videoconferencing component. Group videoconferencing early interventions have been less studied but show promise. Research suggests that they are feasible and yield results comparable to face-to-face interventions, with high participant satisfaction (Banbury et al., 2018).

Although the results of group videoconference interventions have been promising so far, additional research is needed to identify optimal methods of conducting group videoconferencing to maximize clinical benefit and treatment outcomes (Gentry et al., 2019). Additionally, it is essential to carefully consider the adaptation of interventions that were previously conducted face-to-face to the online format, as they may not necessarily have the same acceptability, feasibility and effectiveness (Martin et al., 2020).

In two previous research projects, we developed and evaluated a face-to-face group intervention aimed at promoting maternal sensitivity during a child's first year of life (Figueroa-Leigh et al., 2013; Olhaberry et al., 2020). This intervention was subsequently adapted to a group videoconference format under the name "C@nnected."

The primary aim of this study is to evaluate the feasibility and acceptability of "C@nnected," a group videoconferencing intervention, designed to enhance maternal sensitivity in mother–infant dyads from socially vulnerable areas receiving care in primary health care (PHC) settings in Santiago, Chile. The secondary aims are (1) to identify key parameters for the implementation of the intervention, setting the stage for a larger-scale evaluation of its effectiveness with a randomized control trial (RCT), and (2) to estimate the effect size (ES) of the intervention on maternal sensitivity, postpartum depressive symptoms, postnatal bonding and child socioemotional development by comparing clinical outcomes between groups post-intervention.

## Methods

The protocol describing the methods in detail was previously published (Binda et al., 2022), and the trial was registered on ClinicalTrials.gov (NCT04904861; https://clinicaltrials.gov/show/NCT04904861). This feasibility trial is reported with the Consolidated Standards of Reporting Trials guideline for pilot trials (Eldridge et al., 2016). Local ethical approval was obtained.

### Trial design

The study was conducted from August 2021 to December 2022, being a single-blind, parallel, two-arm, feasibility RCT. Participants were randomly assigned to the intervention group (IG) or the control group (CG) (usual care and educational brochures), in a 3:2 ratio. Participants were blinded to their group assignment. In addition, a nested qualitative study evaluating the perceptions of the providers and intervention users was performed. Figure 1 shows the study process.

### Settings and participants

Participants were recruited from two public PHC centers located in two low socioeconomic level counties in Santiago, Chile: La Pintana y Puente Alto. These counties have levels of poverty, overcrowding and illiteracy higher than the country's average. The study population consisted of mothers with infants aged 6–12 months. The inclusion criteria were (1) being 18 years or older, (2) fluency in Spanish and (3) access to an electronic device capable of videoconferencing. The exclusion criteria were (1) the presence of a severe intellectual disability or current psychotic symptoms and (2) participation in another early intervention program.

### Procedures

Potential study participants were identified from a list provided by each PHC center. Two research assistants (RAs) called the mothers on the list and evaluated the inclusion/exclusion criteria. Those who were interested in participating were sent informed consent online. After signing the electronic consent, the baseline evaluation (t0) and a sociodemographic survey were carried out. The post-intervention

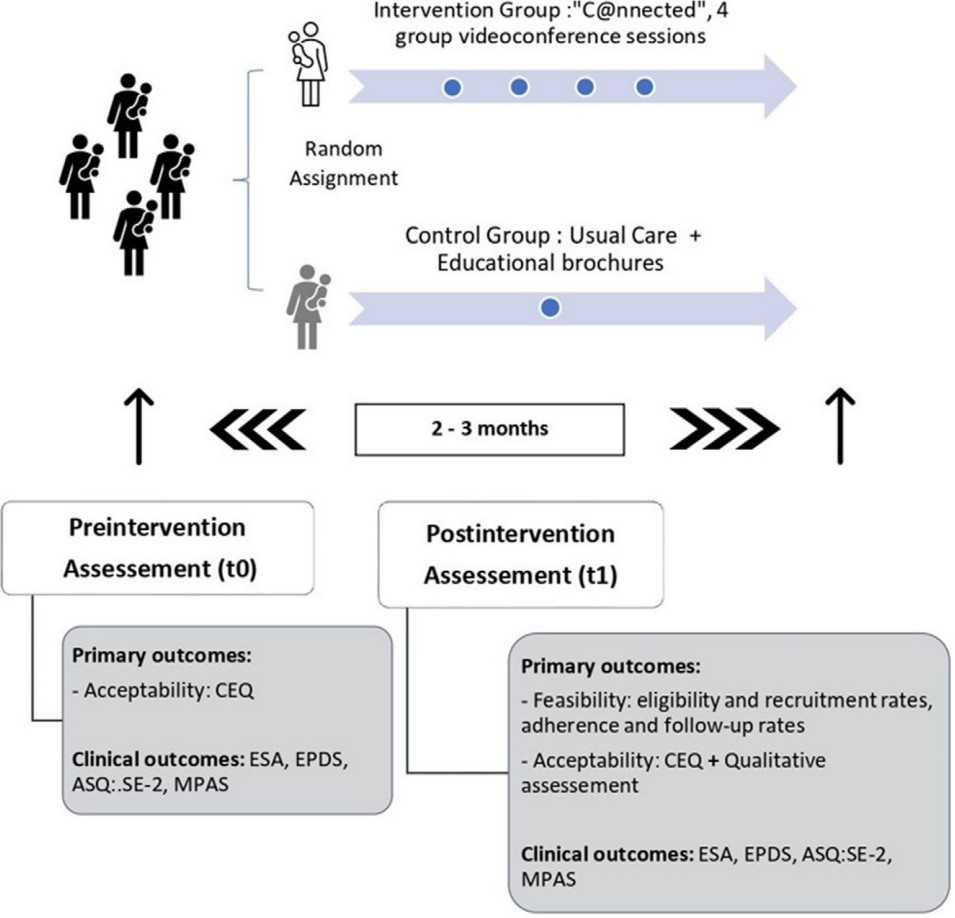

**Figure 1.** Study process. *Note*: ASQ:SE-2: Ages and Stages Questionnaires, Social–Emotional; CEQ: Measurement of Credibility/Expectancy Questionnaire; ESA: Adult Sensibility Scale; EPDS: Edinburgh Postnatal Depression Scale; MPAS: Maternal Postnatal Attachment Scale; TAU: treatment as usual.

assessment (t1) was conducted by the RA 2–3 months after receiving the intervention, repeating the baseline measurements (Figure 1).

### Randomization and masking
Permuted block randomization using a software program (https://www.studyrandomizer.com/) was performed. A block size of 5 was used, in a ratio of 3:2, placing the largest number of participants in the IG. After completing the initial evaluation (t0), the RA was informed of the assignment to which the dyad corresponded. Participants were blinded to the conditions of the two arms. All assessments were performed blindly to the assigned group.

### Outcome measures

#### Primary outcome measures
The primary outcomes measures are feasibility and acceptability.

#### Feasibility parameters
1) Eligibility rates: proportion of mothers who meet inclusion criteria compared with the total number of mothers who were contacted by telephone.
2) Recruitment rates: proportion of mothers who accept the invitation to participate in the study with respect to those who meet eligibility criteria.

3) Adherence to intervention: (a) proportion of participating mothers in the intervention group who receive the intervention and (b) average number of sessions attended.
4) Follow-up rates by treatment condition: proportion of participants who completed the post-intervention assessment.

#### Acceptability parameters
Quantitative assessment.
1) Credibility/Expectancy Questionnaire (CEQ): quantitative satisfaction with the intervention in both groups was measured using the Spanish version of this measure (Devilly and Borkovec, 2000). The term "expectancy" refers to patients' beliefs about how likely they are to benefit from treatment, while credibility refers to how believable, convincing and logical treatment is (Kazdin, 1979). CEQ is a brief and easy-to-administer scale for measuring treatment expectancy and credibility, which has demonstrated good internal consistency (? .79–.86) and reliability. It comprises six items and two factors (credibility and expectancy). Items 1–3 measure credibility (relevance, satisfaction and recommendation), while items 4 and 5 estimate the expectancy (usefulness perception) on a scale of 1–10. Item 6 shows rejection with intervention. Higher scores indicate higher acceptability, except in the rejection item. This measurement was administered before (t0) and after the intervention (t1), to both mothers in the IG and those in the CG.

*Qualitative assessment.* A qualitative assessment was conducted to explore the experiences of both intervention providers and participating mothers, identifying factors associated with acceptability and feasibility, as well as barriers and facilitators to implementation. The evaluation was carried out through:

1) Semi-structured interviews with intervention providers: These interviews explored observed changes, participant achievements, learning outcomes, applied methodologies, implementation challenges, potential improvements and core components of the intervention.
2) Focus groups with participating mothers: Two focus groups, each consisting of four to six mothers, were conducted to assess the perceived usefulness of the intervention, reported achievements, learning experiences, satisfaction levels, applied methodologies and suggestions for improvement.

Data obtained from the qualitative information (the interviews and focus groups) were recorded, transcribed and assigned codes to ensure the anonymization and protection of the participants' identities.

### Secondary outcome measures

Clinical outcome measures were used as secondary outcomes to characterize the sample and provide preliminary estimates of the intervention's effects, which will inform the sample size calculation for a future effectiveness study. Given the small sample size, no significant changes were expected. These measures were assessed at two time points: baseline (t0) and post-intervention (t1), 2–3 months after the completion of the intervention (Figure 1).

The specific measures are as follows:

- **Adult Sensitivity Scale, Escala de Sensibilidad del Adulto (ESA)** (Santelices et al., 2012): Maternal sensitivity was measured using this scale, which assesses an adult's sensitivity in their interaction with children aged between 6 and 36 months. The mother and child are filmed for 5 min during free play interaction. The only instruction given is, "Do what you always do." The coding system considers rubric with 19 indicators that are related to different aspects of the sensitivity response. Each indicator is given a score between 1 and 3, with a higher score indicating higher sensitivity. The ESA gives scores for 3 scales: responsiveness, playful encouragement and warm attunement. The total score for sensitivity, which is calculated by averaging all the items, ranges between 1 and 3. The instrument reliability was .93 based on Cronbach's alpha. For this study, only the total score for sensitivity was considered. The videos were coded by two external blinds researchers. The coders reached a reliability of .675 (Cohen's kappa) for the first 20 videos indicating moderate reliability. This was considered acceptable, as it exceeds the value reported in the original study of the instrument.
- **Edinburgh Postnatal Depression Scale (EPDS)** (Cox et al., 1987): Postpartum depression symptoms were measured using this scale. This scale, which is globally used for the screening of maternal depression (O'Connor et al., 2016), is a self-administered questionnaire of 10 multiple-choice questions. The maximum score is 30, with higher scores indicating higher depressive symptomatology. The scale has been validated in Chile (Jadresic et al., 1995), with adequate internal consistency (Cronbach's alpha = 0.77). We used the cutoff point of ≥10 to consider the presence of postpartum depressive symptoms.
- **Ages and Stages Questionnaires: Social–Emotional (ASQ:SE-2), Spanish Version** (Squires et al., 2015): This questionnaire was used to measure child socioemotional development. The 6- and 12-month versions were applied, depending on the infant's age at the time of evaluation. A higher score indicates poorer socioemotional development, allowing for a direct assessment of the intervention's effects on participating infants. ASQ:SE-2 is a validated tool widely used in early mental health intervention programs (McCrae and Brown, 2018) due to its precision and ease of use in evaluating children's socioemotional development and identifying potential issues, with adequate psychometric properties (internal consistency Cronbach's alpha = 0.82, test–retest reliability 0.94, sensitivity 0.82 and specificity 0.92). It consists of nine caregiver-completed questionnaires addressing seven key behavioral areas: self-regulation, developmental perception, adaptive functioning, autonomy, affect, social communication and interaction. The 6-month version is suitable for ages 3–9 months, whereas the 12-month version is used for ages 9–15 months. Each version includes a specific cutoff score, with infants scoring above the cutoff categorized as "at risk for socioemotional development delay."
- **Maternal Postnatal Attachment Scale (MPAS)** (Condon and Corkindale, 1998): This scale was used to measure postnatal maternal bonding. This self-report measure has 19 items, ranging from 1 (low bonding) to 5 (high bonding). It provides a total score, with a maximum of 95 points, and three subscales: quality of bonding, the absence of hostility and pleasure in interaction. The psychometric properties have shown adequate internal consistency (Cronbach's alpha = 0.78).

### Group videoconferencing intervention: "C@nnected"

This intervention was adapted from a face-to-face brief attachment-based intervention to an e-mental health format aimed at promoting maternal sensitivity in mother–infant dyads (under 12 months) attending PHC. The intervention was previously designed and piloted (Figueroa-Leigh et al., 2013), considering the available evidence and local qualitative information.

The adaptation to the videoconferencing format was carried out between May and July 2021 by two authors of the face-to-face intervention in conjunction with a designer experienced with digital interventions. We used the recommendations of the Early Intervention Foundation (Martin et al., 2020) to achieve good results in virtual interventions as a frame of reference, which includes the use of engagement elements and mechanisms to improve adherence and flexibility. We reviewed the complete content of the workshop, adjusted the sessions from 2 to 1.5 h, and adapted the training manual and all the activities for use in a web-based group environment. We aim to provide lighthearted and interactive activities to allow all participants to understand and share their experiences.

The intervention consists of four videoconference sessions delivered over 4 weeks, each lasting 1.5 h per week. Each group includes three to six dyads led by a trained provider (psychologists). The primary objective of the intervention is to enhance maternal sensitivity and develop skills for interpreting infant cues and responding sensitively. All sessions and activities are conducted with the mother and baby together. Each session focuses on specific themes and objectives, delivered through experiential activities that immediately put the concepts into practice with the baby. In one session, an additional primary caregiver is actively invited to participate. At the end of each session, participants receive educational brochures summarizing key concepts and homework assignments to practice the newly learned skills at home with their families.

The training manual has self-explanatory material regarding each session in order to standardize the intervention. The manual

specified the structure and content of each session, the details of the materials to be used, relevant aspects to be emphasized during the activities and additional information that went more deeply into the issues addressed in the workshop. This seeks to ensure the comprehensive replication of the intervention. Before conducting the intervention, the providers received a half-day training from the authors of the intervention. During this training, providers learn how to carry out each activity and how to be attentive to mothers' sensitive responses throughout the process. For example, one of the activities, "Recognizing Sensitive Responses in Caregivers," involves working with photographs of caregivers interacting with their children in various situations. Mothers are asked to analyze the images and determine whether the caregivers are being sensitive or not, justifying their observations. They are encouraged to reflect on how they can identify sensitivity in these interactions. Table 1 presents the main components of the intervention, along with detailed examples of the activities involved.

### Control group

The CG served as an active comparator, receiving usual care at their PHC center, which included child health checks, materials for early stimulation and the detection and treatment of developmental delays. Additionally, the CG received digital educational brochures with information on early parenting, distributed weekly for 4 weeks. These brochures, which were also provided to the IG after each

session, briefly summarized the key topics covered in each intervention session and included a small task for families to work on together during the week.

### Intervention group

The IG was invited to participate in the "C@nnected" intervention. To improve adherence to the videoconferencing sessions: (1) the group agreed on a day and time to hold them, (2) a telephone chat group was created to remember the session and send brochures, and (3) in the case of nonattendance of a participant to any of the sessions, the provider called her to summarize the session in 15–20 min and encourage participation in the following group sessions. The participant had to attend at least two videoconferencing group sessions plus two phone calls to be considered as an intervention assistant.

### Data analysis

For the quantitative analysis, descriptive statistics were used to summarize the clinical and sociodemographic characteristics of the groups, as well as eligibility rates, recruitment and adherence. Analysis of covariance was applied to estimate the ESs of differences in clinical outcomes between groups at the post-intervention evaluation, adjusting for pre-intervention scores. Data analysis was performed using IBM SPSS Statistics version 27.

**Table 1.** Main components of the intervention: objectives and activities of each session

| Session | Objectives | Examples of activities | Description |
|---|---|---|---|
| 1. "Knowing each other around attachment" | -Achieve group cohesion. <br>-Provide knowledge about attachment. <br>-Recognize expected affective infant's behaviors. | 1) Dynamics of presentation <br>2) Difficulties in motherhood <br>3) Common myths in attachment and parenting | 1) Mothers present their children as if they could speak. <br>2) Mothers recall difficult times in motherhood and how they overcame it. <br>3) Mothers choose cards with some parenting myth, for example, holding children too much is spoiling. They all comment. |
| 2. "What does my baby need?" | -Recognize crying as a signal of communication. <br>-Different expressions of emotions in babies. <br>-The importance of responding to an infant's needs. | 1) Recognizing sensitive responses in caregivers <br>2) Working with different emotions and expressions in babies | 1) Activity with photos of parents interacting with their children and mothers commenting on whether they seem sensitive. <br>2) Activity with different textures, mothers must evaluate their children's emotions and responses. |
| 3. "Massages and agreements in parenting style" (another relevant caregiver is invited) | -Work on parental sensitivity through massage. <br>-Involve another caregiver in the concept of attachment. <br>-Achieving consensus among caregivers on some relevant issues of parenting. | 1) Infant massage with emphasis on sensitive interaction <br>2) Observation of locally made videos with different parenting situations (infant's signs, safe exploration, reading and difficulties in parenting) | 1) The infant massage technique is taught, and then one caregiver performs it on the baby and the other observes how the baby feels, what they liked most, and so on. <br>2) The couples discuss the videos and try to agree on how they would act in different situations. |
| 4. "Boundaries and positive parenting" | -Understand the importance of setting boundaries with respect and love. <br>-Resolve doubts regarding child abuse. <br>-Recognize behaviors associated with positive parenting. <br>-Know the children's rights. | 1) Difficulties in parenting <br>2) Doubts about the proper way to treat children <br>3) Children's rights | 1) Invent a story from an image that shows a difficulty in parenting and the importance of setting boundaries in parenting. <br>2) Mothers write their doubts regarding the appropriate treatment of children and abuse on an anonymous paper. It is left in a basket and then read by the provider who answers each one. <br>3) A card with children's rights is given, and its importance is discussed and reflected upon. |

For the qualitative analysis, interviews and focus groups were analyzed by a psychologist with expertise in qualitative research, using the Framework Analysis Approach (Ritchie and Spencer, 1994) as the conceptual foundation. A content analysis was conducted, guided by the evaluative questions of the study's qualitative component. The analysis involved classifying and grouping the data into broad nodes (analysis categories), followed by the identification of specific details to uncover commonalities, divergences and emerging elements (emerging categories). The information was then condensed with respect to the guiding questions, focusing on participants' perceptions of learning, barriers and facilitators.

## Results

### Feasibility

Participants were enrolled from August 2021 to July 2022; data collection was completed in December 2022. It should be noted that at the beginning of the study, there were important confinement measures in Chile, given by the COVID-19 pandemic, which were progressively decreasing over time. The recruitment and study flow diagram are presented in Figure 2.

The eligibility rate was 89% (122/137). The recruitment rate was 36% (44/122). Thirty mothers declined to participate, citing the

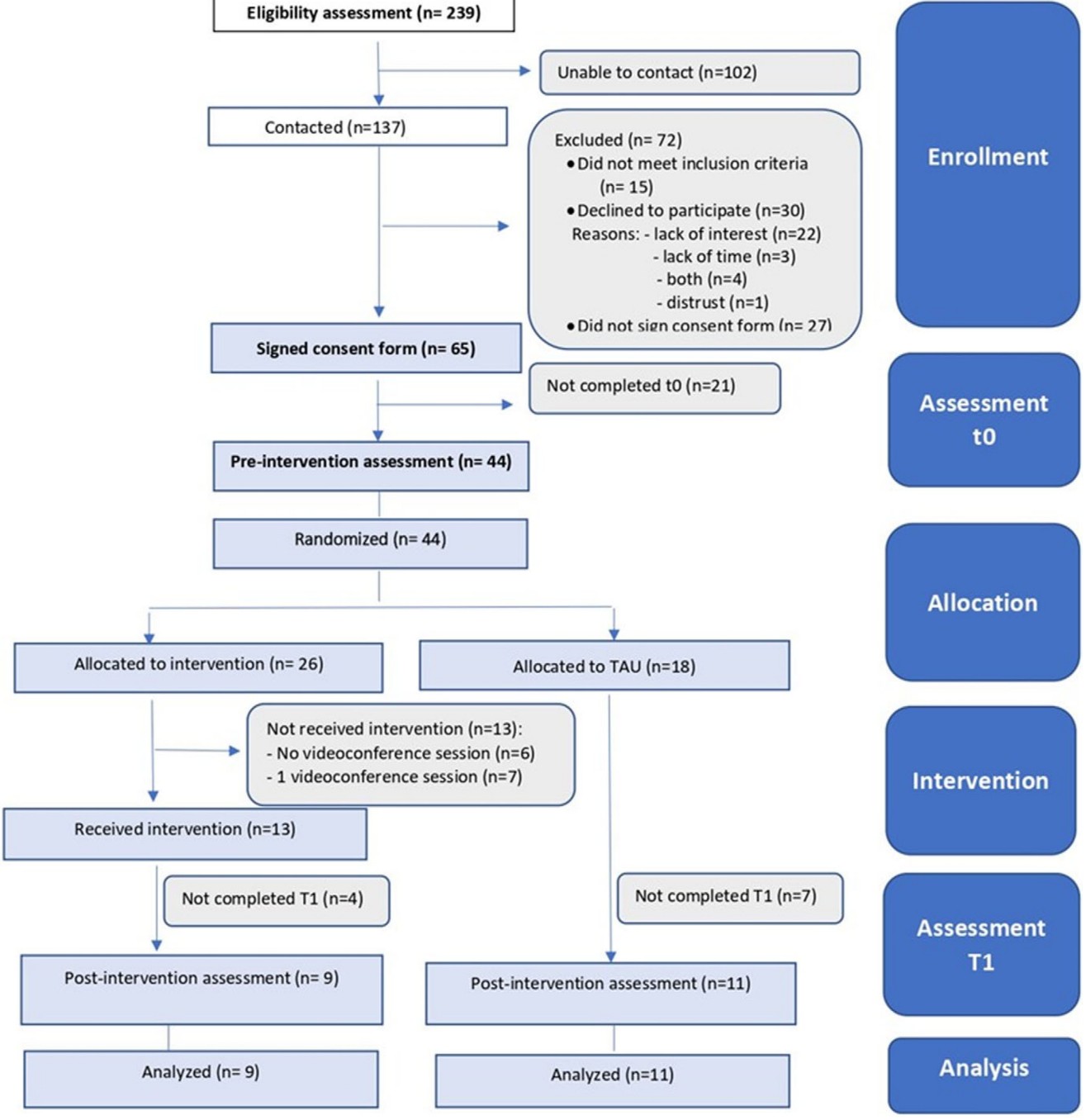

**Figure 2.** Flow diagram. *Note*: TAU, treatment as usual.

main reasons as a lack of interest and insufficient time. The remaining individuals did not enter the study either due to failure to sign the consent form or incomplete baseline measures. Most of those recruited (84%) were from one of the two health centers, which is the one with lower social vulnerability. Twenty-six mothers were randomized to the IG. Table 2 shows demographic characteristics and baseline evaluations of the participants. The average age of the mothers was 29.7 years (SD 5.79), and the average age of the children was 7.55 months (SD 1.42). Notably, 52% of the mothers presented clinical symptoms of postpartum depression evaluated with EPDS.

### Delivery feasibility

Only 13 of the 26 mothers randomized to the IG received the complete intervention (50% adherence to the intervention). Of the mothers who did not receive the intervention, six did not attend any videoconference session and seven only attended one session. In total, five workshops were held by videoconference, between September 2021 and August 2022. A decrease in participation was observed over time coinciding with the reduction of confinement measures in Chile. The average attendance at the sessions was 3.1 sessions (out of 4).

### Follow-up feasibility

The follow-up evaluation was completed at 64.5% (9/13 in the IG and 11/18 in the CG).

### Acceptability

#### Quantitative acceptability

The results are shown in Table 3. All CEQ credibility scores (items 1–3) increased in the IG and decreased in the CG, from baseline to post-intervention, and all the scores were over 9/10 in the IG. The mean difference (MD) between the IG and the CG, after controlling for baseline, ranged from 1.57 to 2.08 points, with a large ES (Cohen's $d$ 0.94–1.36). The same was observed in the expectancy items (items 4 and 5), showing an increase in the IG scores, all above 9/10, and a decrease in the CG. The MD between the IG and the CG was between 1.94 and 3.24, with a large ES (Cohen's $d$ 0.96–1.36). In item 6 (rejection), no changes were observed before and after the intervention in either group.

#### Qualitative acceptability and feasibility

To assess qualitative acceptability and feasibility, the intervention providers were interviewed, and two focus groups were conducted with intervention participants during two distinct implementation periods – at the beginning and end of the COVID-19 pandemic.

The qualitative analysis identified four main categories, offering deeper insights into the implementation process and addressing key evaluation questions: (1) perception of the intervention, (2) facilitators and barriers, (3) implementation requirement and (4) suggestions for improvement. These categories and their subcategories were systematically coded during the analysis and are summarized in Table 4.

**Table 2.** Characteristics of groups at baseline ($N = 44$)

| Characteristics | Intervention group ($n = 26$)<br>$n$ (%)/M + SD | Control group ($n = 18$)<br>$n$ (%)/M + SD | Total ($N = 44$)<br>$N$ (%) | $\chi^2$/t | $p$ |
|---|---|---|---|---|---|
| *Mother* | | | | | |
| Age (years) | 30.31 + 4.87 | 29.00 + 6.98 | 29.77 + 5.79 | 0.73 | .468 |
| Years of education | 14.79 + 3.32 | 14.28 + 3.49 | 14.57 + 3.37 | 0.49 | .63 |
| Chilean Nationality | 25 (96) | 15 (83) | 40 (91) | 2.12 | .146 |
| Paid job | 14 (54) | 8 (44.4) | 22 (50) | 0.38 | .54 |
| Cohabitation with father | 22 (85) | 11 (61) | 33 (75) | 3.13 | .007 |
| Nonplanned pregnancy | 17 (65) | 12 (67) | 29 (66) | 0.01 | .930 |
| Previous history of depression | 18 (69) | 11 (61) | 29 (66) | 0.31 | .576 |
| Depression during pregnancy | 10 (38) | 3 (17) | 13 (30) | 2.43 | .119 |
| *Infant* | | | | | |
| Male | 16 (62) | 10 (56) | 26 (59) | 0.16 | .691 |
| Age (months) | 7.37 + 1.43 | 7.83 + 1.47 | 7.55 + 1.42 | −.12 | .269 |
| Fist child | 9 (35) | 8 (44) | 17 (39) | 0.16 | .691 |
| *Clinical outcomes* | | | | | |
| Total maternal sensitivity (ESA) | 1.96 + .42 | 2.00 + .46 | 1.98+ .43 | −.31 | .728 |
| Low maternal sensitivity (ESA) | 7 (27) | 6 (33) | 13 (30) | 1.70 | .428 |
| EPSD score | 11.35 + 7.14 | 10.22 + 7.03 | 10.89 + 7.04 | 0.52 | .608 |
| Postpartum depressive symptoms | 14 (54) | 9 (50) | 23 (52) | 0.06 | .802 |
| MPAS total score | 79.5 + 6.34 | 78.81 + 7.98 | 79.09 + 7.28 | −.31 | .76 |
| ASQ:SE total | 40.28 + 22.19 | 37.69 + 36.64 | 38.75 + 31.26 | −.27 | .791 |

*Note*: ESA, Escala de Sensibilidad del Adulto (Adult Sensibility Scale); EPDS, Edinburgh Postnatal Depression Scale; MPAS, Maternal Postnatal Attachment Scale; ASQ:SE-2, Ages and Stages Questionnaires: Social–Emotional.

**Table 3.** Quantitative acceptability measured using the Credibility/Expectancy Questionnaire

| | Intervention group (*n* = 9) | | | Control group (*n* = 11) | | | | |
| | Pre-intervention *M* (SD) | Post-intervention *M* (SD) | X^ | Pre-intervention *M* (SD) | Post-intervention *M* (SD) | X^ | Diff. IG-CG | D |
|---|---|---|---|---|---|---|---|---|
| *Credibility* | | | | | | | | |
| 1. Relevance | 8.22 (2.64) | 9.11 (1.69) | 8.96 | 7.64 (2.16) | 7.27 (2.15) | 7.39 | 1.57 | 0.94 |
| 2, Satisfaction | 8.33 (2.69) | 9.44 (1.01) | 9.41 | 8.00 (2.24) | 7.55 (1.86) | 7.58 | 1.83 | 1.22 |
| 3. Recommendation | 8.67 (2.65) | 9.67 (.50) | 9.64 | 8.45 (2.30) | 7.55 (1.07) | 7.57 | 2.08 | 1.36 |
| *Expectancy* | | | | | | | | |
| 4. Usefulness 1 | 7.89 (2.47) | 9.56 (.88) | 9.58 | 8.18 (2.18) | 6.36 (3.04) | 6.34 | 3.24 | 1.36 |
| 5. Usefulness 2 | 8.89 (1.35) | 9.33 (1.32) | 9.32 | 8.82 (1.72) | 7.36 (2.54) | 7.38 | 1.94 | 0.96 |
| *Rejection* | | | | | | | | |
| 6. Rejection | 1.89 (2.98) | 1.67 (3.20) | 1.63 | 1.73 (3.35) | 1.64 (2.46) | 1.66 | −0.03 | 0.01 |

*Note*: X^, mean post-intervention controlling for mean pre-intervention; CEQ, Credibility/Expectancy Questionnaire.

1) **Perception of the Intervention**: Participants reported high levels of satisfaction with the intervention. Mothers highlighted increased knowledge, improved bonding with their children and a greater sense of competence and fulfillment in their maternal role.
2) **Facilitators and Barriers:**
   a) Facilitators: Key facilitators included the structured and organized nature of the intervention, the use of a detailed manual and supporting materials and activities that promoted reflection and discussion among participants. Additionally, the experience, flexibility and commitment of the intervention providers played a crucial role in the program's success.
   b) Barriers: Participants expressed mixed opinions regarding the online modality. While some preferred to continue virtually, others expressed a desire to return to in-person sessions as health restrictions eased. Additional barriers included maternity leave and competing responsibilities, such as caring for other children or managing work commitments.
3) **Implementation Requirements:**
   a) Physical Resources: A stable internet connection and, ideally, access to devices with larger screens (like personal computers) were recommended for better visualization of materials and group interactions. However, most mothers primarily used smartphones due to their availability, ease of use and accessibility.
   b) Provider Skills: Essential skills for intervention providers included active listening, group management, moderation of discussion spaces, improvisation, problem-solving, tolerance for frustration, flexibility, teamwork and empathy.
4) **Suggestions for Improvement:** Participants suggested expanding certain topics within the intervention, such as postpartum depression, the role of play in child development and increasing father involvement.

### Clinical outcomes

The results showing changes in clinical outcomes before and after the intervention are presented in Table 5 and Figure 3.

### Maternal sensitivity

Total maternal sensitivity scores increased slightly, both in the CG and the IG. No differences were found in the change in total maternal sensitivity between groups evaluated with the total ESA scale.

### Maternal depression

The mean EPDS score in the IG decreased, whereas, in the CG, it remained unchanged. When comparing groups, the MD between them shows that the IG had a greater reduction in depressive symptoms compared with the CG (not statistically significant) with a large ES (*d* = 0.82). The mean post-intervention EPDS score, after controlling for the pre-intervention score, was 9.14 in the IG, which is below the clinical threshold for maternal depression, whereas, in the CG, it was 12.56, which remains above the cutoff point.

### Socioemotional development

ASQ:SE scores increased slightly, both in the CG and the IG in the post-intervention assessment. No differences were found in the change between groups.

### Postnatal maternal bonding

An increase in the MPAS total score was observed in both the IG and the CG. When comparing groups, the MD between them shows that the IG had a greater increase in maternal bonding compared with the CG (not statistically significant.) with a medium ES (*d* = 0.6). The mean post-intervention MPAS total score after controlling for the pre-intervention score was 83.2 in the IG out of a maximum of 95 points.

### Discussion

The present study sought to evaluate the feasibility and acceptability of "C@nnected," a group videoconference early intervention aimed to enhance maternal sensitivity in mother–child dyads served in PHC in Chile. Both quantitative and qualitative methodologies were used to explore the subjective perspective of the participants regarding the intervention.

In terms of feasibility, a high eligibility rate (89%) was obtained, but a lower recruitment rate (36%) was observed, which reflects a

**Table 4.** Qualitative analysis

| | | | |
|---|---|---|---|
| Facilitators and Barriers | Facilitators | Structure and organization of tde material | "Personally, I like tdese kinds of metdodologies (…) Give your opinion, share ideas, reflect (…) I find tdat tdey were very playful." FG 1; P1. |
| | | Training and professional experience | |
| | | Monitor flexibility and commitment | |
| | | Material delivered to tde participants | |
| | | Reflective activities and space for conversation | |
| | | Motders' motivation to learn | |
| | Barriers | Unstable Internet connection | "I think when they are in school it's easier, because (…) Now my daughter is at school and I am alone with the baby and I can do the activities with you, but maybe if I had more children (…) I could not be 100% in the workshop (…)." FG 2; P4. |
| | | Device used not optimal for viewing (cell phone) | |
| | | Change in sanitary condition | |
| | | Ambivalence toward the online modality vs. need for face-to-face accompaniment | |
| | | End of the postnatal leave period | |
| | | Other demands of the mother that make it difficult to dedicate exclusively to the workshop | |
| Perception of the intervention | Perception of the workshop | Motivation to learn and bond with their children | "I feel that these workshops serve a lot (…) To accompany you, to know other opinions, to reflect, to diminish a little the burden of what it is to be a mother (…) then these workshops serve a lot, more than anything for the accompaniment of the mother." FG1; P3. |
| | | Expansion of knowledge and correction of previous erroneous ideas | |
| | | Activities that promote interaction, reflection and sharing of experiences | |
| | | Appropriate and easy to perform tasks | |
| | | Improvement in knowledge, communication and contact with their children | |
| | | Feeling of greater competence and satisfaction in the role of mother | |
| | | Recognition of the importance of asking for help and sharing experiences with other mothers | |
| | Perception of the delivered material | High satisfaction | "At night (…) I reviewed the material, I read it, it was very good (…) I showed it to my partner (…) and he read it, he also informed himself (…)." FG1; P2. |
| | | Learning facilitator | |
| | | Possibility to share | |
| Requirements for implementation | Resources | Good Internet connection and the availability of electronic devices (ideally PCs or tablets) | "I believe that today we all have access to the internet (…) normally in Santiago the connection is good (…)." FG 2; P3. |
| | | Quality audiovisual material | |
| | | Exclusive dedication time to the workshop | |
| | Monitor skills | Soft skills (active listening and containment) | "(…) The psychologist was very playful, (…), had good material, if you didn't understand her, she explained to you, or sometimes nobody wanted to talk, and she took the initiative (…) and gave you the confidence to speak (…)." FG1; P2. |
| | | Need for supervision spaces | |
| | | Clinical experience | |
| Suggestions for improvement | Dig deeper into the topics addressed | | "I think that more than a topic, it could also be a workshop for fathers… Postpartum depression is an issue (…)." FG1; P1. |
| | Include other topics: postpartum depression, play with children, paternal involvement | | |
| | Modification of order and duration of activities and topics | | |

*Note*: FG, focus group; P, participant.

significant need for mental health care and, at the same time, challenges in recruiting for remote group intervention. These results may be explained by the fact that the recruiters were external to the PHC center and unknown to the mothers, which may have negatively impacted motivation and trust in the intervention. Additionally, mothers who agreed to participate had lower psychosocial vulnerability than those who declined to participate in the intervention. This allows us to hypothesize that living conditions without privacy spaces, inadequate technological resources and family contexts with more stressors, present in families with higher psychosocial vulnerability, could inhibit participation in this type of intervention. Considering these results, for a future effectiveness

**Table 5.** Clinical outcomes

| | Intervention group (*n* = 9) | | | Control group (*n* = 11) | | | | |
|---|---|---|---|---|---|---|---|---|
| | Pre-intervention *M* (SD) | Post-intervention *M* (SD) | X^ | Pre-intervention *M* (SD) | Post-intervention *M* (SD) | X^ | Diff. IG-CG | *D* |
| *ESA* | | | | | | | | |
| Total | 2.20 (.45) | 2.32 (.40) | 2.32 | 2.22 (.46) | 2.37 (.32) | 2.37 | −0.06 | 0.15 |
| Empathic response | 1.98 (.57) | 2.17 (.43) | 2.18 | 2.04 (.62) | 2.26 (.32) | 2.26 | −0.08 | 0.22 |
| Playful interaction | 2.20 (.53) | 2.39 (.42) | 2.39 | 2.23 (.44) | 2.48 (.44) | 2.48 | −0.10 | 0.22 |
| Emotional expression | 2.44 (.33) | 2.41 (.49) | 2.40 | 2.41 (.45) | 2.39 (.37) | 2.40 | 0.00 | 0.00 |
| *EPDS* | 11.67 (7.28) | 8.44 (6.25) | 9.14 | 13.64 (6.59) | 13.09 (5.75) | 12.56 | −3.39 | 0.82 |
| *ASQ:SE* | 7.38 (6.64) | 9.19 (6.15) | 9.66 | 9.30 (6.08) | 10.33 (6.05) | 9.14 | −0.27 | 0.05 |
| *MPAS* | | | | | | | | |
| Total | 80.0 (5.0) | 84.0 (3.0) | 83.25 | 77.64 (6.8) | 78.82 (9.11) | 79.43 | 3.82 | 0.60 |
| Bonding quality | 40.56 (3.91) | 43.33 (1.5) | 43.07 | 39.18 (2.99) | 39.82 (4.05) | 40.03 | 3.04 | 1.00 |
| Non-hostility | 16.56 (1.51) | 17.11 (1.83) | 17.15 | 16.64 (2.77) | 16.64 (3.5) | 16.61 | 0.54 | 0.25 |
| Pleasure | 22.89 (2.15) | 23.56 (1.81) | 23.21 | 21.82 (3.31) | 22.36 (2.66) | 22.64 | 0.57 | 0.34 |

*Note*: X^, mean post-intervention controlling for mean pre-intervention; ESA, Adult Sensitivity Scale; ASQ:SE, Ages and Stages Questionnaires: Social–Emotional; EPDS, Edinburgh Postnatal Depression Scale; MPAS, Maternal Postnatal Attachment Scale.

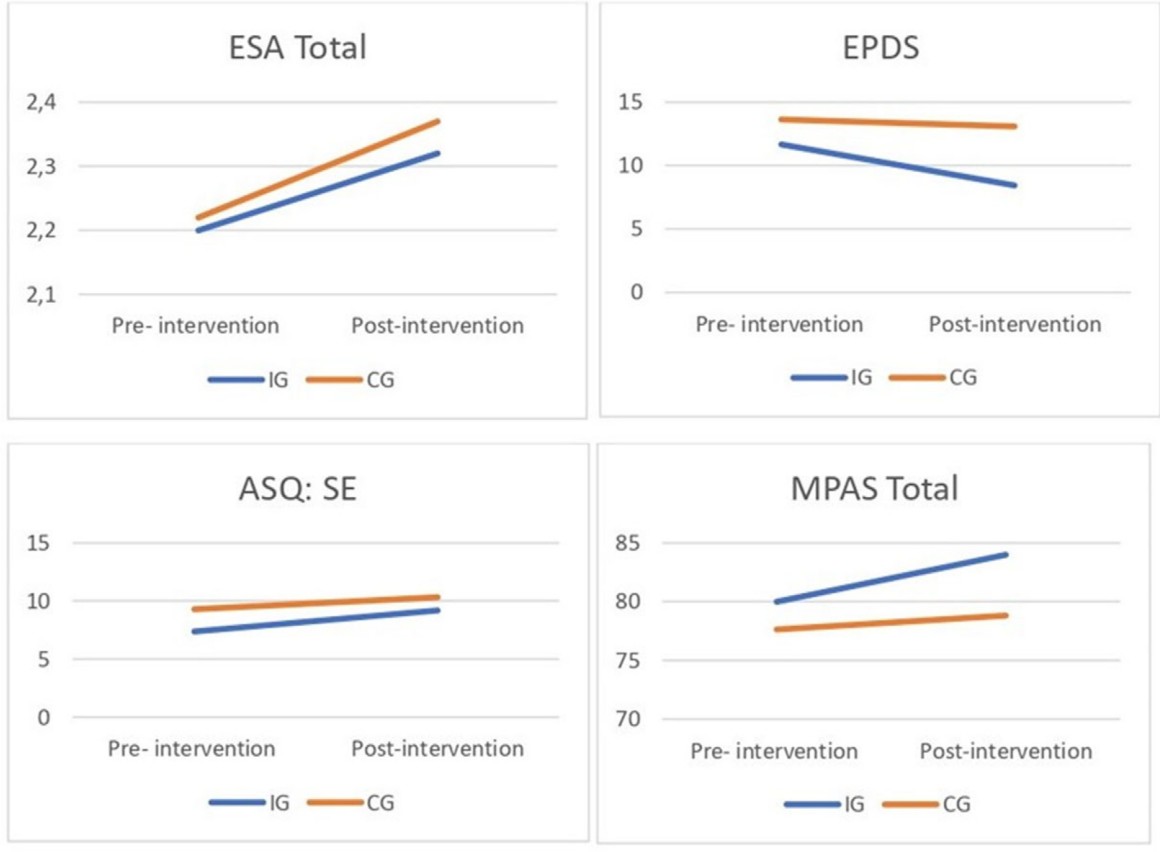

**Figure 3.** Changes in clinical outcomes. *Note*: ESA, Adult Sensitivity Scale; ASQ:SE, Ages and Stages Questionnaires: Social–Emotional; EPDS, Edinburgh Postnatal Depression Scale; MPAS, Maternal Postnatal Attachment Scale; IG, intervention group; CG, control group.

study of the developed intervention, it is proposed to use recruiters from the corresponding PHC center and to consider mother–infant dyads with low or moderate psychosocial vulnerability. These adjustments would ensure the necessary conditions for participation and contribute to the motivation and trust of participants, based on prior contact with the person inviting and establishing the

first connection. In this regard, it is possible that the developed intervention may not meet the needs of higher-risk psychosocial groups, requiring other modalities to address their mental health care needs.

While evidence shows that parental interventions have been effective (Niccols, 2008) and up to six times less expensive than individual interventions (Bunting, 2004), they report low adherence, close to 30%, as their main difficulty (Patterson et al., 2002). The adherence to the intervention conducted was 50%, which is higher than the adherence reported in in-person group interventions for parents, which is particularly relevant considering its importance and the challenges in achieving it, as reported in previous studies. Most recent studies on online parenting programs for early childhood show promising results; however, they are primarily self-guided, and therefore, adherence to these programs cannot be directly extrapolated to those delivered via videoconferencing (Kempe et al., 2025). A feasibility study of an attachment-based program adapted for videoconferencing reported adherence rates of approximately 50% (Labella et al., 2023), similar to what was observed in this study. In this study, a decrease in adherence was observed as health restrictions eased, so it is necessary to consider that the adherence obtained cannot necessarily be extrapolated to post-pandemic conditions. Another factor to consider in adherence is postpartum maternal depression, present in more than half of our sample, which could negatively affect adherence to the intervention, especially considering its group modality.

The acceptability of the intervention, quantitatively assessed using the CEQ instrument, was excellent, with scores nearing the maximum after the intervention on both credibility and expectancy items. Credibility refers to the believability, persuasiveness and logical consistency of the intervention, while outcome expectancy reflects participants' beliefs about the likelihood of benefiting from the intervention (Kazdin, 1979). Credibility is thought to be shaped by a logical process, whereas outcome expectancy arises from a more affective process (Devilly and Borkovec, 2000). Empirical evidence indicates that outcome expectancy more frequently predicts treatment outcomes than credibility (Thompson-Hollands et al., 2014). Both constructs should be assessed, especially in studies involving novel interventions. The high levels of both credibility and expectancy observed in this study suggest strong participant satisfaction with the intervention, which may also indicate the potential for positive clinical outcomes.

Through the qualitative evaluation, we examined the acceptability, feasibility, perceived benefits of the intervention and key implementation factors from the perspectives of both beneficiary mothers and intervention providers. The data collected reflect consistently positive perceptions from both groups, particularly in relation to increased parenting knowledge, enhanced mother–infant bonding and greater competence and satisfaction in the maternal role. These outcomes are closely aligned with the intervention's intended objectives and anticipated impact. Recent studies have similarly reported that mothers participating in online parenting interventions perceive significant gains in knowledge and parenting skills (Shorey and Ng, 2019; Feil et al., 2020).

Participants also expressed a strong appreciation for the internal dynamics of the workshop. The relevance and appropriateness of the topics were widely recognized, with a particular emphasis on the value of reflective and dialogic activities. These components were seen as instrumental in fostering a collaborative and experiential learning environment that enriched interactions among mothers and supported their need for social connection. Previous research has underscored the effectiveness of active learning strategies in promoting participant engagement and knowledge acquisition (Smith et al., 2018; Gomez et al., 2022).

Overall, both facilitators and participating mothers reported high levels of satisfaction and a strong sense of the workshop's usefulness – findings that are consistent with the quantitative results of this study.

Regarding the feasibility of the intervention, participants expressed positive views about the materials provided, the structure and organization of the sessions, the flexibility of the program and the professional qualities of the facilitators. Similar findings have been reported in the literature. A recent scoping review of early childhood parenting programs highlighted the importance of a clear program structure, adequate resources and facilitator competence as key elements supporting program effectiveness (Silva et al., 2022). These converging findings reinforce the idea that both logistical and relational aspects of program delivery are critical to the feasibility and overall acceptability of parenting interventions.

The external context – specifically the mobility restrictions during the early stages of the COVID-19 pandemic – affected participants' perceptions of the intervention's acceptability, particularly in relation to the online format and barriers to participation and retention. This became evident as perceptions shifted with the gradual return to in-person activities and changes in family dynamics, addressing their need for social support. However, both mothers and facilitators perceived that these elements had a greater impact in in-person settings. The qualitative evaluation also identified key facilitators and barriers in the implementation of the intervention. Facilitators included the structured design of the intervention, the use of the manual and materials and the activities that encouraged reflection and discussion. Additionally, the commitment, experience and flexibility of the intervention providers were crucial factors contributing to their success. These findings emphasize the importance of having well-prepared facilitators and thoughtfully designed materials for effective session delivery. Regarding barriers, not all participants were fully satisfied with the online modality, although it was appreciated when in-person care was not an option. Other barriers included the conclusion of maternity leave, which required some mothers to return to work, limiting their availability to participate, as well as the added challenge of managing care for other children without external support. These facilitators and barriers provide a foundation for a larger-scale evaluation of the intervention's effectiveness, particularly through an RCT.

In relation to the clinical outcomes, they were included as secondary outcomes in this feasibility study to characterize the sample and provide preliminary estimates of the intervention's effects. Given the sample size, significant differences were not anticipated, nor was the study designed to evaluate the intervention's effectiveness. Nevertheless, some of the descriptive results are promising and align with the qualitative findings, particularly the observed changes in maternal bonding and depressive symptoms, suggesting their potential relevance for future effectiveness studies. Conversely, the lack of changes in maternal sensitivity and socio-emotional development highlights the importance of conducting a full-scale effectiveness trial.

The increase in the average postnatal bond observed in the participants of the experimental group is consistent with the findings from the qualitative analysis, in which improvements in the bond with their children were reported. The MPAS, a self-report scale used to evaluate and characterize the sample in relation to the quality of the bond, is easy and quick to use, unlike observational scales for assessing sensitivity, which require recordings and

significant training. Therefore, the MPAS could be an appropriate evaluation tool for use in PHC.

The observed reduction in maternal depressive symptoms, together with qualitative findings, underscores the benefits of sharing motherhood experiences with peers and strengthening maternal competence and satisfaction among mothers with postpartum depression. These results suggest that the intervention may contribute to alleviating mood symptoms. If its effectiveness in reducing depressive symptoms is confirmed in future studies, the intervention could be considered a complementary approach to postpartum depression treatment. This is particularly relevant, given the high prevalence of postpartum depression, especially in Chile, and its well-documented impact on child development.

The study did not show improvements in maternal sensitivity, which may be attributed to several factors. The first is the small sample size. Another potential explanation is the use of a maternal sensitivity evaluation scale developed exclusively in Chile, which lacks robust evidence of its psychometric properties. The moderate reliability level obtained for this assessment may have also influenced the study results. This scale was selected due to its lower coding cost compared with other scales with stronger empirical support, such as the Emotional Availability Scales or the Maternal Behavior Q-Sort or CARE-Index (Bohr et al., 2018). For future studies, it is recommended to use alternative scales with more substantial evidence of validation to ensure more reliable and accurate measurements of maternal sensitivity. Another possible reason for the lack of improvements in maternal sensitivity is the timing of the evaluation, carried out only 2 or 3 months after the intervention. This time period may not have been adequate to observe changes in mother–child interaction, but it was possible to detect better bonding between the mother and her child, which could be the precursor to improvements in sensitivity and interaction in the longer term. It is also possible that the intervention may not lead to significant improvements in maternal sensitivity, considering its brief duration and primarily psychoeducational nature. This highlights the need for an RCT to rigorously evaluate its effectiveness in promoting maternal sensitivity and other parenting outcomes.

In this study, infants' socioemotional development did not show improvement. The potential reasons are similar to those previously mentioned: a small sample size and closely timed follow-up evaluations, which may have limited the ability to detect early improvements in this outcome. For future studies, a delayed assessment is recommended to better capture potential advancements in maternal sensitivity and infants' socioemotional development.

It is important to acknowledge that group-based parenting interventions may not meet the needs of all caregivers. While some participants benefit from the in-person components that promote interaction and foster a sense of social support, others prefer remote formats, as reflected in the qualitative findings of our study. For low-risk populations, the literature has shown the benefits of individual, self-guided online interventions, which are also more cost-effective (Kempe et al., 2025). In contrast, more complex cases may require longer-term interventions with an interdisciplinary approach and individualized care. "C@nnected" was designed as a low-cost, scalable and easily trainable intervention, suitable for implementation in primary health care settings. However, it is likely most appropriate for families at intermediate levels of psychosocial risk. In more complex situations, more intensive formats – such as extended in-person sessions using advanced techniques like video feedback – may be necessary to support meaningful change. A promising strategy to enhance the effectiveness of parenting interventions at the population level is to offer a range of formats tailored to the varying needs, complexities and preferences of caregivers.

## Strengths and limitations

The study combines quantitative and qualitative methodologies, which are recommended for feasibility studies. The intervention used is low cost, easy to replicate, locally created and culturally appropriate for this population and improves access to early parenting interventions.

Some limitations of the study should be considered: (1) The study population comes from highly vulnerable social areas, which may limit the generalizability of the findings to different settings. (2) The study was conducted during the COVID-19 pandemic, with various health restrictions in place, making it uncertain whether the observed feasibility would be similar in different sanitary contexts. (3) The early timing of outcome evaluation (2–3 months postintervention) may have contributed to the lack of changes in some measures. (4) Maternal anxiety symptoms, which are known to influence both maternal sensitivity and child outcomes, were not assessed. (5) The study used a maternal sensitivity evaluation scale with limited psychometric validation and moderate reliability, which may have impacted the measurement of this outcome.

## Conclusion

In summary, the present study shows high acceptability of the intervention quantitatively and qualitatively. Feasibility was lower, primarily due to difficulties in recruitment and adherence intervention, but similar to other in-person group parental interventions. Barriers, facilitators, requirements and suggestions for the implementation of the intervention were detected, which must be incorporated in future research or for implementation on a larger scale.

For future research, an RCT is required to evaluate its effectiveness, incorporating the elements found in this pilot study.

It would be interesting to test the effectiveness of offering different formats of early parenting interventions – group, individual, in-person and remote – tailored to parents' preferences or needs, in order to ensure access for all families to early parental interventions that promote maternal and infant mental health.

### Abbreviations

| | |
|---|---|
| ASQ:SE-2 | Ages and Stages Questionnaires: Social–Emotional |
| CEQ | Credibility/Expectancy Questionnaire |
| CG | control group |
| EPDS | Edinburgh Postnatal Depression Scale |
| ES | effect size |
| ESA | Adult Sensitivity Scale |
| IG | intervention group |
| MD | mean difference |
| MPAS | Maternal Postnatal Attachment Scale |
| PHC | primary health care |
| RA | research assistant |
| RCT | randomized controlled trial |

**Open peer review.** To view the open peer review materials for this article, please visit http://doi.org/10.1017/gmh.2025.10036.

**Data availability statement.** The data that support the findings of this study are available from the corresponding author upon reasonable request.

**Acknowledgements.** The authors gratefully acknowledge the mothers and infants who participated in this study, as well as Madre Teresa de Calcuta and Juan Pablo II Primary Health Care Centers, where this project was conducted.

**Author contribution.** V.B. and M.O. lead the conceptualization of the study and the development of the intervention. The study implementation was overseen by V.B. and coordinated by C.Caa. and C.A. The outreach of participants and intervention implementation were conducted by C.Caa. and C.A. Data analysis was conducted by J.M.-K. All authors contributed to the critical review and interpretation of the findings. V.B. led the drafting of the manuscript, with support from C.Cas. and M.O. All authors critically reviewed and provided feedback on the manuscript.

**Financial support.** This study was funded by the Vicerrectoría de Investigación de la Pontificia Universidad Católica de Chile through the "Concurso de Investigación Interdisciplinaria Convocatoria 2020" and supported by the Millennium Institute for Research on Depression and Personality (MIDAP).

**Competing interests.** The authors declare none.

**Ethical standard.** Ethical approval was obtained from the Scientific Ethics Committee of Health Science of Pontificia Universidad Católica de Chile (#200813008).

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
