## [Reviewer Report]

Group Video Conferencing Intervention “C@nnected” to Enhance Maternal Sensitivity in Primary Care in Chile: Randomized Feasibility Trial

The study reports a complex intervention whose feasibility has been evaluated at many levels. However, the report is very brief and needs more relevant information. My comments and questions aim to help the authors complete the information required to properly evaluate the study’s implementation and findings.

Abstract

Line 15 in the abstract (and line 150 in the method)  implies that the next step in the C@nnected evaluation is an effectiveness study. However, since the current study is a feasibility trial, a randomized controlled trial should be expected to examine the program’s efficacy before testing its effectiveness.

Introduction

The introduction’s first paragraph needs to be updated with more recent studies on postpartum depression rates and their relations with maternal sensitivity. On line 46, the authors affirm, “This study shows Chile as the country evaluated with the highest prevalence of postpartum.” However, the study was mentioned inside the parenthesis. To maintain the wording of the sentence, the study cited in the previous sentence must be out of the parenthesis.

It is crucial to define and briefly discuss predictors of the main variables investigated in the study, such as maternal postpartum depression, maternal sensitivity, maternal bonding, and infant socioemotional development. Discussing in-person and remote interventions to promote maternal sensitivity and reduce maternal postpartum depression is also essential.

Method

Although there is a figure summarizing the measures and procedures, the section about the data collection procedures must cite all the measures and how they were administered (e.g., did mothers self-report postpartum depression and maternal sensitivity?).

The Adult Sensitivity Scale must be more precisely described. What are the sensitivity dimensions or factors evaluated by the scale? How did coders register maternal sensitivity along the 5-minute footage? Is there only a global rating for the whole video, or is it divided into intervals? What criterion was adopted to accept such a low interobserver reliability coefficient (0.675)?

The Ages and Stages Questionnaires must be more precisely described, including the behaviors or skills evaluated, how these behaviors or skills are evaluated (are they self-reported by the mother?), and the measure’s psychometric parameters. Authors affirmed having used “the 6- and 12-month versions”. Were infants evaluated when they were completing 6 or 12 months of age? How did this criterion meet the time (t1) criterion, which required that t1 evaluation occur only 2 or 3 months after intervention completion?

What are the three subscales of the Maternal postnatal attachment scale? This information is essential for readers to understand what maternal bonding is.

The intervention description in the text and table needs to clarify how the dyads' presence during the virtual meetings was explored. Which techniques or strategies were used to practice sensitivity during the sections? Or was the presence of the dyads encouraged only for feasibility reasons (mothers could not leave the infants with other caregivers during the videoconferences)?

The method must include a clear description of the focal group scripts (topics orienting the discussion) and qualitative interviews, as well as a description of the procedure adopted for data analysis (how the authors formulate and implement the analytical categories mentioned in the results section).

Using ANCOVA with small samples requires the attainment of many statistical assumptions regarding data distribution and equality of variance for each assessed variable. The authors should confirm that all the assumptions were tested for all the investigated variables.

Discussion

The intervention reduced maternal depression and improved maternal bonding, both remarkable findings for a short and low-cost online intervention. However, the lack of improvement in maternal sensitivity was attributed to “the use of a sensitivity evaluation scale used exclusively in Chile, which lacks sufficient evidence on its psychometric properties.” This limitation was properly addressed in the discussion. Nevertheless, systematic reviews and meta-analyses show that efficacious interventions aimed at maternal sensitivity usually require more time and direct training of maternal sensitive behaviors using techniques such as video feedback or direct feedback of the observed dyad’s interaction. Another intervention characteristic pointed out in the meta-analysis is having a specific focus. Since c@nnected also covers other essential topics, such as child abuse and difficulties in parenting, time for addressing maternal sensitivity could have been insufficient. I also could not understand if the group videoconference structure allowed interventionists to observe and guide mothers in responding promptly and sensitively to their infants during the group sessions. Maybe this is a severe limitation of online group sessions that must be addressed in the discussion. This is an intervention feature that must be properly discussed.

In sum, this study showed that c@nnected is feasible and has the potential to impact maternal mental health and bonding positively. These two findings must be further investigated in a randomized controlled trial to test the intervention’s efficacy.

Proofreading

The whole text must go through proofreading. There are missing periods, misuse of punctuation, and some incomplete sentences.

---

## [Reviewer Report]

This is an article addressing a globally relevant topic in public policy, applicable to countries with varying characteristics.

In the introduction, the research problem is clearly outlined, aligning well with the study presented, and the authors use updated and pertinent references. I would only suggest adding a brief mention that during the perinatal period, depression and anxiety are the most common pathologies, and this study focuses on depression. Including the primary reason for this decision would be helpful.

Additionally, I suggest adding a brief justification for conducting a feasibility trial beyond what is already indicated in the introduction. It is evident that telematic interventions have increased and shown effectiveness, but why is it necessary to develop models to maximize effects? Isn’t the existing knowledge sufficient?

The methodology is clearly explained, and the included figure is important for understanding the flowchart.

Regarding the instruments, the cited ASQ-SE 2 reference seems to belong to ASQ-SE 1, not version 2. Please verify.

In the intervention, they mention that it was piloted in a 2013 study by Figueroa and refers to the adaptation made for telematic application. In this process, it seems important to provide a more detailed explanation of the adaptation carried out and whether any aspect of the adaptation was piloted before its implementation. For instance, did they adapt physical materials into digital format?

I believe the clinical outcomes of the intervention might confuse readers, as the study was not primarily focused on effectiveness but on feasibility. While the attractiveness of effectiveness results and the outcomes obtained are understandable, I suggest linking these results to specific components of the intervention.

In the case of non-significant results for maternal sensitivity, which is one of the main focuses of the intervention, the discussion about the scale and its psychometric validity is important. However, in the methods section, you report a reliability rate close to acceptable, suggesting that observational instruments inherently face such challenges and that reliability is a central element for reporting trustworthy results.

Considering the other outcomes reported, they could refer to specific components of the intervention and the scale itself, assessing whether there are aspects of the intervention that the scale does not capture and vice versa.

When discussing adherence and noting that not all parenting interventions work equally well for everyone, it seems relevant to consider maternal depression itself as a key factor. Depression may inhibit participation in group formats and interventions that require social interaction. In this sense, the telematic format may have been a protective factor for adherence, as they note it was higher than face-to-face interventions.

In summary, I believe this study deserves to be published, but it should place greater emphasis on its feasibility and acceptability (than effectiveness).

---

## [Reviewer Report]

The paper on “Group Video Conferencing Intervention “C@nnected” to Enhance Maternal Sensitivity in Primary Care in Chile: Randomized Feasibility Trial” shows the challenges and opportunities of group-based online parenting support in Chile. The feasibility trial has some great features such as the pre-registration of the trial, the protocollization of the intervention, the use of observation in addition to the usual self-reports, and the combination of quantitative and qualitative components. Relatively little work in this direction has been conducted in Chile and in South America in general, so it is a welcome addition to the global literature.

Some problems with the current report might be addressed in a revision to make the work more impactful and valid.

First, it is critical to present and describe the study as a feasibility study and avoid any suggestion of a randomized trial that would have anything to tell about the effectiveness of the intervention, and by implication its cost effectiveness. This is ‘just’ a pilot study, an important one from which several lessons can be learnt for a really well-powered RCT but it cannot tell us anything about effectiveness.

Second, in the same line the authors should refrain from any statistics about effectiveness. At the end of the trial only 9 experimental participants and 11 controls remain. The power of a complete case analysis is way too low to draw any conclusion about (absence of) effects. Moreover, intent-to-treat should be the preferred statistical approach but has not been mentioned. Leaving out the section on so-called clinical outcomes would make the paper much more focused on its core messages and be way more convincing.

Third, the sensitivity observational measure is based on only 5 minutes free play. The authors might be a bit more critical of this measures and suggest alternatives such as the MBQS that has been used in other South American countries such as Colombia and is certainly more valid. In Chile some teams have a lot of experience with observational measures for parenting and for attachment. Bonding and other self-reports on attachment have been shown not to be valid assessments of the rather complex attachment construct and in a large trial the authors should take the opportunity to use such more time-intensive but also more rewarding approaches.

In sum, the reported study is impressive but only as a feasibility study.

---

## [Reviewer Report]

The manuscript has been improved, and only a few details still need revision:

Page 4, line 66 to line 76: Clarify to which countries or regions the prevalences of post-partum/perinatal depression and anxiety refer to.

Page 5, line 116: The authors use “telematic” without defining it. A brief definition can be inserted in brackets.

Page 8, line 252: The moderate reliability level obtained for maternal sensitivity assessment must be discussed as a possible limitation that may have impacted the study results regarding the intervention effects.

Table 1: The intervention curriculum is clear. However, the intervention activities or strategies were not fully described. More transparency is required in reporting how providers acted during the vídeo conference to perform activities such as “Recognizing sensitive responses in caregivers,” “Working with different emotions and expressions in babies,” or “Infant massage with emphasis on sensitive interaction.” Did providers merely offer verbal instructions on how to “recognize sensitive responses,” “work with different emotions,” or “sensitively massage the baby,” or were there specific strategies to help mothers share their own experiences and discuss them?

Table 3: This table shows that participants in the control group also answered the CEQ. However, it is unclear what it refers to: Does the CEQ for the control group refer only to the brochures, or does it refer to all the “business as usual” treatments at the PHC centers? This information must be inserted in the measure description or in the procedures section. The table must display F, DF, and p values from between groups and within groups ANCOVA.

Table 5: The table must display F, DF, and p values from between groups and within groups ANCOVA.

Page 12—Qualitative acceptability: The quality of writing in this section is low. There are truncated sentences, and punctuation was misused.

Page 15, lines 526-542: Sentences from lines 526 to 533 are repeated in lines 534-542.

There are many grammar errors and/or typos in the Whole manuscript. There are truncated sentences, and punctuation was misused in many sections. Thus, the whole manuscript must go through proofreading.

---

## [Reviewer Report]

The authors did not take into account most of my review of the original submission, in particular the following points:

First, it is critical to present and describe the study as a feasibility study and avoid any suggestion of a randomized trial that would have anything to tell about the effectiveness of the intervention, and by implication its cost effectiveness. This is ‘just’ a pilot study, an important one from which several lessons can be learnt for a really well-powered RCT but it cannot tell us anything about effectiveness.

Second, in the same line the authors should refrain from any statistics about effectiveness. At the end of the trial only 9 experimental participants and 11 controls remain. The power of a complete case analysis is way too low to draw any conclusion about (absence of) effects. Moreover, intent-to-treat should be the preferred statistical approach but has not been mentioned.

Computing effect sizes without CIs is misleading as are suggestions about cost-effectiveness without knowing what the program’s effectiveness is.

Really, the authors should refrain from inferential statistics, and any suggestion of effectiveness or efficacy, and just present their feasibility study. With lots of lessons to be learnt about invalid measures etc.

---

## [Editor Report]

You will see that a reviewer has raised major concerns about your inappropriate use of inferential statistics and over generalization from a very small sample.

I am prepared to offer a final opportunity to revise the manuscript in line with the reviewer’s recommendations and if these changes are not made then with regret I will have to decline publication.

---

## [Reviewer Report]

The manuscript was improved, and only a few adjustments are needed:

This sentence in the introduction (line 105) needs revision (the excerpt “more evidence is needed” should be excluded): "Although the results of telematic group videoconference interventions have been promising so far, additional research is needed to identify optimal methods of conducting group videoconferencing to maximize clinical benefit and treatment outcomes, more evidence is needed (Gentry et al., 2019).

In the discussion section (line 530), regarding retention rates, the authors cite only one study by Tuntipuchitanon et al. (2022), which compared an online group-based parenting intervention to business as usual in a highly educated sample. Currently, other empirical studies and reviews reporting on adherence and retention rates in online parenting programs could be explored, especially in the previous paragraph (line 505). The article by Kempe et al. (2024) shows some of these studies (Kempe, S., Barriault, S., Lovegrove, A., Laverdiere, S., & Racine, N. (2025). Making connections matter: Enhancing child development through online parenting interventions. Practice Innovations, 10(1), 90–101. https://doi.org/10.1037/pri0000258). Additionally, the study by Tuntipuchitanon et al. (2022) was retracted and updated in 2024. Thus, the citation must be updated as well.

From line 567 to line 607, the text resembles more of a results section than a proper discussion section. I suggest that the authors search the literature for insights to discuss these findings.

The intervention has many strengths and has proven feasible, with high acceptability and participant satisfaction. There is also evidence of its effectiveness in reducing maternal depression and enhancing maternal bonding, which is crucial to positive parenting. Nevertheless, the two group sessions exclusively focused on maternal sensitivity are insufficient to benefit both maternal sensitivity and infant development. First, showing pictures of mother-infant dyads and asking participants to share their thoughts on sensitivity is a very interesting technique to approach the sensitivity issue, but it is unlikely that it will change participants' behaviors towards their infants. Second, the focus on the massage (the baby is passive) does not allow the facilitator to encourage mothers to respond to the infant’s initiatives, which is a core feature of sensitivity. This limitation must be acknowledged and discussed for future online and in-person parenting interventions.

The Whole manuscript must go through proofreading to improve grammar and avoid typos and misspelling.

---

## [Reviewer Report]

TThere are still two minor issues to be revised:

The conclusion must be updated according to the changes implemented in the discussion. For example, the last sentence emphasizes the intervention’s impact on maternal sensitivity, whereas the intervention outcomes were detected on maternal depression (mental health) and bonding:

“It would be interesting to test the effectiveness of offering different alternatives of parenting interventions, depending on the parents' preferences or complexity: group, individual, in- person and remote, in order to guarantee access for all families to parental interventions that enhance their parental sensitivity.”

The verb to be in the following sentence should be conjugated in the past tense (substitute “are” by “were”) in the following excerpt:

“(...) requirements and suggestions for the implementation of the intervention are detected (...).”